# Learning to Rank for Active Learning via Multi-Task Bilevel Optimization

Zixin Ding[1]        Si Chen[2]        Ruoxi Jia[2]        Yuxin Chen[1]

[1]Department of Computer Science, University of Chicago, Chicago, Illinois, USA
[2]Department of Computer Science, Virginia Tech, Blacksburg, Virginia, USA

## Abstract

Active learning is a promising paradigm for reducing labeling costs by strategically requesting labels to improve model performance. However, existing active learning methods often rely on expensive acquisition functions, extensive model retraining, and multiple rounds of interaction with annotators. To address these limitations, we propose a novel approach for active learning, which aims to select batches of unlabeled instances through a *learned surrogate model* for data acquisition. A key challenge in this approach is to develop an acquisition function that generalizes well, as the history of data, which forms part of the utility function's input, grows over time. Our novel algorithmic contribution is a multi-task bilevel optimization framework that predicts the relative utility—measured by the validation accuracy—of different training sets, and ensures the learned acquisition function generalizes effectively. For cases where validation accuracy is expensive to evaluate, we introduce efficient interpolation-based surrogate models to estimate the utility function, reducing the evaluation cost. We demonstrate the performance of our approach through extensive experiments on standard active classification benchmarks.

## 1 INTRODUCTION

Many decision making tasks involve maximization of utility functions [Chen et al., 2015b, Jackson et al., 2019]. As an example, utility in active learning (AL) can be represented in various forms, such as expected error rate reduction [Mussmann et al., 2022, Roy and McCallum, 2001], mutual information between the labeled and unlabeled datasets [Sourati et al., 2016, Adaimi and Thomaz, 2019, Lindley, 1956], or the uncertainty of model predictions [Settles, 2012, Shen

et al., 2017, Kossen et al., 2022]. However, maximizing utility under budget constraints in AL is notoriously challenging. It is well-known that determining the optimal set containing maximal information under cardinality constraint is NP-hard [Ko et al., 1995, Chen et al., 2015a]. In classification tasks, determining the groundtruth utility of *subset* of training data needs retraining classifier on that set (and then evaluate it on the validation set). It's computationally infeasible to calculate out the utility for the *best* possible subset for downstream tasks without carefully examining every possible subset [Engstrom et al., 2024]. Moreover, common AL methods rely on acquisition functions with high *adaptivity* to the environment, in which the selection choices for current round depend on the responses to the labeling requests for all previous rounds. This reliance poses major concerns for the deployment of these algorithms to real-world applications, as there could be a substantial delay between requesting labels and receiving feedback. For instance, in scientific experiments, feedback from wet-lab or physics experiments can take days or even months to obtain [Botu and Ramprasad, 2015, Yang et al., 2019], limiting the rounds of interactions with labelers, thus bearing the risk of sampling redundant or less effective training examples within a batch.

We thus ask: *How to develop a robust acquisition criterion for AL with only one round of interaction with annotators given fixed budget constraints?* So far, dominant AL approaches rely on customized utility metrics characterizing the current model's behavior. Recent works [Ash et al., 2019, Killamsetty et al., 2021, Saran et al., 2023, Sener and Savarese, 2017] propose to use gradients of the current model based on the *pseudo labels* of the unlabeled data. Yet, these gradient estimates can be unreliable for single round AL setting due to the limited data of labeled pool. The datamodels framework [Ilyas et al., 2022] showcases the linear relationship between training data and model predictions, a seemingly promising paradigm for designing acquisition function. It is worth noting that the framework is under *supervised* settings, i.e. requiring *labeled* subsets of train-

ing data and studies how the choice of training set affect model predictions. Conversely, acquisition criteria in AL are defined as function mapping from *unlabeled* instances, or instances without label information, to real utility value.

In this paper, we focus on enhancing the *robustness* and *generalizability* of deep active learning under one round setting. Given the variability in deep learning models due to different initializations, hyperparameters, network architectures and training procedures [Jiang et al., 2021, D'Amour et al., 2022, Zhong et al., 2021], the one-shot estimate of validation accuracy can be highly stochastic, and thus we resort to the idea of *ranking* as a strategy to mitigate the inherent uncertainty. Rather than learning a predictor for validation accuracy, we shift the perspective towards (approximately) comparing which subset of unlabeled pool would lead to better generalization on validation set. Concretely, in Section 4, we aim to *approximate* the relative utility value of equal size subset of training data via a novel variant of RankNet [Burges et al., 2005], which we refer to as the *utility model*. This is achieved by integrating a set-based neural network architecture, enabling us to extend comparisons from individual *examples* to pairs of *sets*.

To accommodate the increasing labeled pool, we separate samples based on the size of inputs and employ bilevel training to account for the growing training history. We introduce a multi-task learning framework that uses the optimal transport distance [Alvarez-Melis and Fusi, 2020] between the current labeled data and validation set as an additional loss, regularizing the utility model to enhance generalization to new, unlabeled data, while being agnostic to training dynamics of the underlying classifier. Furthermore, to refine the utility model estimation and reduce the computational overhead of obtaining groundtruth utility samples during the pretraining stage, we employ interpolation-based techniques to augment utility samples (defined in Section 4.1).

We summarize the above algorithmic insights into a novel learning-based acquisition strategy, namely RAMBO (Ranking-based Active learning via Multitask Bilevel Optimization), as illustrated in Fig. 1. We conducted extensive experiments on various active learning benchmarks in image classification, and showed that RAMBO consistently outperforms existing learning/regression-based active learning algorithms by a significant margin. Our method offers a promising alternative for maximizing data utility under budget constraints, unlocking potential applications in a wide range of classification tasks.

## 2 RELATED WORK

**Utility model learning.** Surrogate models build on a rich and growing body of machine learning literature [Konyushkova et al., 2017, Coleman et al., 2019, Kossen et al., 2022, Ilyas et al., 2022, Wang et al., 2023, Engstrom

et al., 2024]. These works improve data acquisition by: training a regressor to predict expected error rate reduction [Konyushkova et al., 2017, Wang et al., 2023], trimming down model architectures/training epochs as proxy models [Coleman et al., 2019, Wang et al., 2023], approximating the distribution of labels and unobserved features [Li and Oliva, 2021], and harnessing datamodels [Ilyas et al., 2022] framework by minimizing trained model loss on target tasks [Engstrom et al., 2024]. In contrast, our method use ranking-based neural networks to *learn* a data acquisition function(utility model) *estimating* which subset would yield higher utility value given a pair of equal size of subset training data. Contrary to existing works, we train the utility model by collecting much less samples and sampling from various sizes of subsets rather than fixing subset sizes.

**Planning-based vs learning-based AL strategies.** Classical AL have predefined acquisition strategy including uncertainty sampling [Settles, 2012, Shen et al., 2017, Gal et al., 2017], diversity sampling [Sener and Savarese, 2017, Yehuda et al., 2022] or their combined approaches [Xie et al., 2022, Citovsky et al., 2021, Parvaneh et al., 2022]. Meanwhile, there is a long line of work on *learning-based* acquisition function [Fang et al., 2017, Bachman et al., 2017, Wang et al., 2023, Sinha et al., 2019, Yan et al., 2022, Yoo and Kweon, 2019, Li and Oliva, 2021, Killamsetty et al., 2021]. For instance, priors works combine meta-learning [Killamsetty et al., 2021] or semi-supervised learning [Borsos et al., 2021] with bi-level optimization in designing acquisition functions. Yoo and Kweon [2019] adopt the idea of ranking the predicted classifier loss in comparing two instances as "loss prediction module", querying instances that the classifier is likely to predict wrong, and learn it to predict target losses of unlabeled inputs. We draw inspirations from Killamsetty et al. [2021], Yoo and Kweon [2019] by leveraging bi-level training as a subroutine for enhancing generalizability of utility model and querying highest ranked unlabeled instances.

**Learning to rank.** Ranking techniques have been foundational in fields such as information retrieval [Liu et al., 2009], recommendation systems [Karatzoglou et al., 2013, Li, 2022] and large language models [Ouyang et al., 2022]. Motivated by Yoo and Kweon [2019], Li et al. [2021], we shift from the traditional approach of learning cross-entropy loss on unlabeled instances to ranking the utility for paired subsets of data. While both works [Yoo and Kweon, 2019, Li et al., 2021] view ranking predicted losses as an uncertainty measure, our methodology centers on gauging the utility of labeled data subsets, with the utility being the validation accuracy post-training. To the best of our knowledge, our method is the first to incorporate the idea of ranking between pairs of subsets and link it directly to the performance of the learning algorithm on the validation set. We will show the computational advantages and empirical successes of integrating RankNet [Burges et al., 2005] and sidestepping

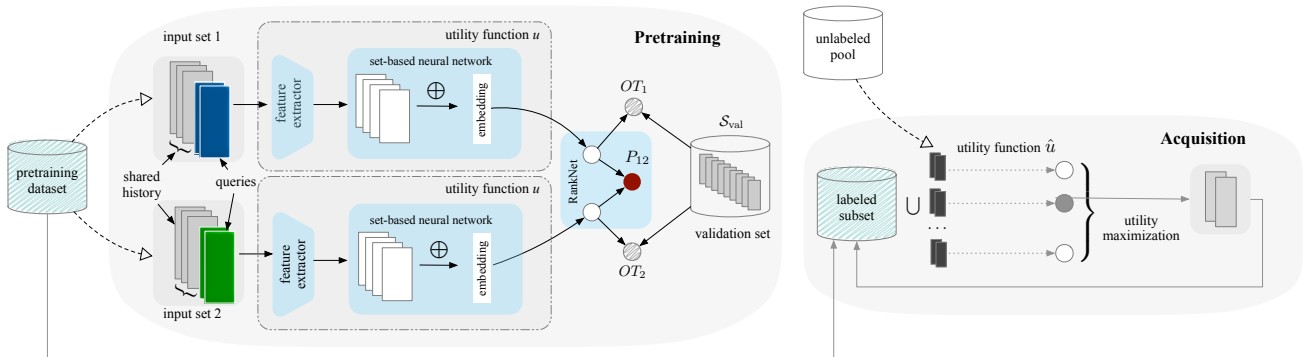

Figure 1: Overview of the RAMBO algorithm. For pretraining stage, we learn a RankNet over pairs of utility samples via multi-task bilevel optimization; for acquisition stage, we follow the learned utility function to iteratively query data points in minibatches. Details of the algorithm are provided in Section 4.

regressing on unlabeled subsets in Section 5.4.

## 3   PROBLEM STATEMENT

Consider a ground set of data points $\mathcal{X}$ with the groundtruth labeling function $f^* : \mathcal{X} \to \mathcal{Y}$. The active learning problem in our study unfolds in a two-stage protocol:

1. A *pretraining stage*, where we train an acquisition function from an initial pool of data points;

2. An *acquisition stage*, where we actively select a set of new examples to label all at once.

We denote the initial pretraining (labeled) set by $\mathcal{S}_0$ with $\mathcal{S}_0 \subseteq \mathcal{X}$ and $|\mathcal{S}_0| = k$, and denote the labeled set after the acquisition stage by $\mathcal{S}_1$ with $|\mathcal{S}_1| = k + B$ where $B$ represents the labeling budget. The unlabeled sets before and after acquisition are represented as $\mathcal{U}_0$ and $\mathcal{U}_1$ respectively. The groundtruth utility function is defined as $u : 2^{\mathcal{X}} \to \mathbb{R}$, where $u(\xi)$ quantifies the utility of a subset $\xi \subseteq \mathcal{X}$ by evaluating the validation accuracy of the classifier $f$ induced by the (labeled) data in $\xi$. Our goal is to find the optimal subset $\mathcal{S}_1^*$ such that $f$, when trained on it, achieves the highest validation accuracy, thereby optimizing the utility function $u$:

$$\mathcal{S}_1^* \in \underset{\mathcal{S}_0 \subseteq \mathcal{S}_1 \subseteq \mathcal{X}, |\mathcal{S}_1 \setminus \mathcal{S}_0| = B}{\arg\max} u(\mathcal{S}_1) \qquad (1)$$

Here, $u(\mathcal{S}_1) = \mathbb{E}_x[\mathbb{1}(f(x) \neq f^*(x)) \mid \mathcal{S}_1]$ for classification tasks, and can be estimated by the error rate of the resulting $f$ on a validation set $\mathcal{S}_{\text{val}} \subseteq \mathcal{X}$.

Learning $u$ in Equation 1 is challenging in our setting. Indeed, even approximating $u$ requires the groundtruth utility for a large collection of subsets of the labeled pool, under the practical constraints of a limited labeling budget. We emphasize that the instances are selected *non-adaptively* in the acquisition stage, i.e., our selection of instances does not depend on the labels of previously selected instances.

We aim to devise an acquisition strategy for subset selection with maximal downstream classification accuracy.

## 4   METHODOLOGY

We introduce our algorithm, RAMBO, following the two-stage learning protocol described previously. In a nutshell, RAMBO (1) collects training samples for *pretraining* utility model, and (2) greedily selects the batch with the maximal estimated utility value from one to total batches $t$ in the acquisition stage. We divide the pretraining stage into $\tau_1$ iterations and the acquisition stage into $\tau_2$ iterations with mini-batch size $b$ for each iteration. More precisely, we instantiate RAMBO into the following building blocks:

a) Develop a set-based multitask neural network model $\hat{u}$ as a surrogate model for pertaining;

b) Define the loss function for the utility model $\hat{u}$;

c) Sample a collection of subsets $\{(\xi, u(\xi))\}_i \subseteq \mathcal{S}_0$ where $i \in [1, \tau_1]$ as a growing labeled set up to $\mathcal{S}_0$ for training $\hat{u}$;

d) Update the set based model $\hat{u}$ per iteration of the pre-training stage;

e) Greedily follow the learned utility model $\hat{u}$ in the acquisition stage.

### 4.1   A TWO-STAGE ACTIVE LEARNING FRAMEWORK

So far, we have defined the framework and will unravel a)-d) above to discuss each relevant aspect respectively:

**a) What surrogate models $\hat{u}$ should we use?** Similar to Ilyas et al. [2022], by parametrizing a surrogate model with training samples, we transform the surrogate model construction into a supervised learning task (See Definition 1). In our context, the training samples $\xi$ are subsets of pretraining set $\mathcal{S}_0$ and the utility value is $u(\xi)$. Throughout this work,

we refer to the pairs $(\xi, u(\xi))$ as *utility samples*. It is appealing to adopt their linearity assumption into AL setting due to strong theoretical footing [Saunshi et al., 2022] and simplicity in model architectures. Nevertheless, to avoid extensive sample collection and model retraining [Ilyas et al., 2022], we hence prefer more complex architectures for modeling the interaction between elements within each utility sample. One natural candidate for $\hat{u}$ is set-based neural networks due to their strong expressive power (i.e., Set Transformer [Lee et al., 2019] or Deep Sets [Zaheer et al., 2017]). Denote the general set-based neural network(NN) as

$$\text{net}(\xi) = \text{net}(x_1, ..., x_a) = \rho(\text{pool}(\{\phi(x_1), ...\phi(x_a)\}))$$

where $\{x_i\}_{i=1}^a$ represents a single utility sample $\xi$ with size $a$ and $\phi, \rho$ is the feature extractor and regressor for the set-based NN itself.

In experiments, we find set-based NN shall serve as a primitive for utility models, but still, it lacks principled supervision signals for model training. Engstrom et al. [2024] train millions of cheap datamodels [Ilyas et al., 2022] in the hope of better generalization for unseen tasks, while in our setting, we shall not afford large-scale training due to computational infeasibility and aim to obtain good utility model with hundreds of samples for faster deployment. Therefore, we need a more fine-grained signal that would tie labeled data and validation set. In particular, Alvarez-Melis and Fusi [2020] introduce the notion of geometric distance via optimal transport (OT) between two datasets and Just et al. [2023] extend it as a learning-agnostic proxy for measuring model performance on $\mathcal{S}_{val}$. The celebrated success of OT distance in predicting validation set accuracy [Just et al., 2023] enables us to cast the groundtruth OT distance between utility samples and validation set [Alvarez-Melis and Fusi, 2020] as a supervision signal for the utility model.

**Definition 1** (Surrogate Utility Model). *Let $\mathcal{X}$ be the instance domain, and $\xi$ be any sampled subset drawn from the distribution $\mathcal{D}$ over $\mathcal{X}$. A surrogate utility model $\hat{u}(\xi)$ is a set function mapping from $2^{\mathcal{X}} \to \mathbb{R}$, optimized to predict the true utility $u(\xi)$ on a training set $\xi \sim \mathcal{D}$:*

$$\hat{u} = \underset{\tilde{u}_w}{\arg\min} \, \hat{\mathbb{E}}_{\xi \sim \mathcal{D}}[\mathcal{L}(\tilde{u}_w(\xi), u(\xi)] \tag{2}$$

*where $\mathcal{L}(\cdot, \cdot)$ denotes the loss function, and $\tilde{u}_w$ is a parametric set function to approximate $u$.*

**b) What loss function should we minimize?** One natural choice is to minimize the MSE (mean square error) of estimated and true utility value as $\mathcal{L} = (\hat{u} - u)^2$. Yet, the evaluation of validation accuracy is non-deterministic (thus stochastic) due to the aleatoric uncertainty of the classifier itself [Park et al., 2023]. While the simplistic way is to train a neural network to approximate the utility value in a regression fashion and minimize the MSE, we fail to learn a good utility model by regressing validation accuracy on a set of

utility samples (See Section 5.4 for ablation study on casting utility model as regression). An alternative for loss function lies in the idea of pairwise ranking, simplifying regression problem to ranking problem. Yoo and Kweon [2019] introduce a loss prediction module to predict the classifier loss on a single data point and handicraft the loss function for predicting the classifier loss in a pairwise ranking fashion. For minibatch samples with size $d$, Yoo and Kweon [2019] divide it into $d/2$ pairs and rank the differences between each pair of predicted and groundtruth losses to discard the overall loss scale. Extending the idea of ranking classification loss between pairs of instances to rank the utility value, we incorporate the classical RankNet [Burges et al., 2005] structure to rank between pairs of equal-size utility samples with OT distance as a regularizer in the final loss.

**Definition 2** (Ranking Loss). *Given $\mathcal{X}$, and let $\xi_1, \xi_2$ be two sampled subset drawn from distribution $\mathcal{D}$ over $\mathcal{X}$ with equal size $d$. Denote the utility value (validation accuracy) of $\xi_1$ as $u_1$ and the utility value of $\xi_2$ as $u_2$. W.l.o.g. suppose $u_1 > u_2$, $u_{12} = u_1 - u_2$. Specifically, $u_1 > u_2$ is taken to mean that the surrogate utility model $\hat{u}$ asserts that $\xi_1 \rhd \xi_2$. Denote the modeled posterior $P(u_1 \rhd u_2)$ by $P_{12}$, and let $\bar{P}_{12}$ be the desired target values for those posteriors. The Binary Cross Entropy (BCE) loss for pair $(\xi_1, \xi_2)$ is written as*

$$\mathcal{L}_{Rank} = -\bar{P}_{12} \log P_{12} - (1 - \bar{P}_{12}) \log(1 - P_{12}).$$

With this metric in hand, we shall guide $\hat{u}$ to learn the principled signal ties to validation set accuracy and ignore the shifting distribution between labeled data and $\mathcal{S}_{val}$ in the acquisition stage (See Definition 3). Even though OT distance can be approximated in near-linear time complexity [Altschuler et al., 2017], our goal is to estimate which subset of training data yields the highest validation accuracy rather than approximating OT distance itself. To circumvent the computational infeasibility, we leverage the OT distance as a supervision signal to regularize $\hat{u}$ rather than serving as an input to $\hat{u}$. We show the efficacy of incorporating OT Distance Loss in Section 5.4.

**Definition 3** (OT Distance Loss). *Given two utility samples $\xi_1, \xi_2$. Denote the corresponding ground truth OT distance values by $OT_1, OT_2$, and the predicted OT distance values as $\hat{OT}_1$ and $\hat{OT}_2$. The OT distance loss is defined as*

$$\mathcal{L}_{OT} = \lambda_1(\hat{OT}_1 - OT_1)^2 + \lambda_2(\hat{OT}_2 - OT_2)^2$$
$$- \lambda_3(\min(\hat{OT}_1, 0) + \min(\hat{OT}_2, 0))$$

*where $\lambda_1, \lambda_2, \lambda_3$ are hyperparameters.*

Here, the first two terms are mean squared error for OT distances and the third terms are positive constraints. Intuitively, the OT distance loss specifies the penalty for mispredicting the OT distance values of utility samples $\xi_1$ and $\xi_2$. Combining the ranking loss and the OT distance loss, we obtain the loss function for RAMBO over pairs of utility samples:

**Definition 4** (Total Loss for Utility Model). *Given two utility samples $\xi_1, \xi_2$. The total loss over $(\xi_1, \xi_2)$ is defined as*

$$\mathcal{L}_{Total} = \mathcal{L}_{Rank} + \lambda_{OT} \cdot \mathcal{L}_{OT} \tag{3}$$

*where $\lambda_{OT}$ is a hyperparameter.*

**c) How do we collect utility samples iteratively?** The very first question encountered during pretraining is how to generate utility samples. Ilyas et al. [2022] construct training subsets by random sampling a fixed-length subset. One caveat in our setting is the growing length of labeled sets as the progression of the active learner. To enable the model to adapt to the growing length of utility samples, one needs to incorporate *diversity* in the size of $\xi$. One natural choice is to perform rejection sampling from the *powerset* of $\mathcal{S}_0$, i.e., $\xi \sim 2^{\mathcal{S}_0}$. Instead of fixing the sampling proportion, we propose to fix the number of utility samples collected from $\mathcal{S}_0$ per iteration during pretraining as $n$.

**d) How do we update the set-based NN during pretraining?** As mentioned in Section 4.1, the length of labeled utility samples grows, and random split for training and validation set may fail to capture the notion of generalizability in neural batch active learning. The goal of the utility model is to *generalize* to the longer length of utility samples and learn a general mapping from utility sample to validation accuracy. Inspired by bilevel training work [Franceschi et al., 2018, Grazzi et al., 2020, Borsos et al., 2021], we employ a bilevel framework to separate the utility samples by *length*. In practice, we separate the validation set and training set by 50% and 50% for simplicity. We retrain the set-based NN per iteration with the accumulation of utility samples per iteration. We defer the complete discussion of bi-level training to Section 4.2.1.

**e) How do we acquire data in the acquisition stage?** In the context of utility maximization, perhaps the simplest candidate is to select the instance with the largest predicted utility. Popular approaches rely on sequentially picking one data point per round [Houlsby et al., 2011, Gal et al., 2017] though the addition of a single data point causes minimal change to validation accuracy while increasing the cost of model retraining. Alieva et al. [2020] suggest that for many sequential decision making problems, greedy heuristics for sequentially selecting actions exhibit superior performance without invoking expensive evaluation oracles. Recall that one shall interpret $\hat{u}$ as a score-based acquisition function and leverage it for sequential decision making, i.e. to greedily select unlabeled data with the highest predicted utility. Inspired by Citovsky et al. [2021], we employ Margin Sampling [Roth and Small, 2006] as a filter for unlabeled instances i.e., select $M$ unlabeled instances with lowest margin scores, per iteration in the acquisition stage (See Algorithm 2). We propose to randomly split $\mathcal{U}_0$ into batches of size $b$, concatenate each batch to the current labeled pool,

---

**Algorithm 1** RAMBO

1: **Input**: $B, \mathcal{U}_0, \mathcal{S}_0 \; \mathcal{X}, b, M, n, \mathcal{S}_{val}$.
2: **Output**: $\mathcal{S}_1$
3: Initialize $(\hat{u}_0, \phi_0)$ from offline dataset
4: Randomly divide $\mathcal{S}_0$ with size $k$ into $S_0$ with size $k_1$ and $\{s_1, s_2...s_{\tau_1}\}$ with each size $b$ and set $U_0 = \mathcal{U}_0$
5: $\tau_1 = \frac{k-k_1}{b}$ and $\tau_2 = \frac{B}{b}$
6: Train $f$ on $S_0$ and get accuracy on $\mathcal{S}_{val}$ as $acc_0$
7: $\mathcal{D}_0 \leftarrow \{\}$
8: **for** $i = 0 : \tau_1$ **do**                          ▷ **Pretraining**
9:      $S_{i+1} \leftarrow S_i \cup \{s_{i+1}\}$
10:     Train $f$ on $S_{i+1}$
11:     Obtain accuracy on $\mathcal{S}_{val}$ as $acc_{i+1}$
12:     $D_{i+1} \leftarrow$ Utility-Samples-Augmentation$(S_i,$
13: $S_{i+1}, n, acc_i, acc_{i+1}, D_i)$
14:     Train $\hat{u}_i$ from $D_{i+1}$        ▷ **Bilevel Optimization**
15: **for** $j = 0 : \tau_2$ **do**                          ▷ **Acquisition**
16:     $S_{j+1}, U_{j+1} \leftarrow$ Greedy-Margin$(\hat{u}_{\tau_1}, j, b, S_j, M, U_j)$
17: $\mathcal{S}_1, \mathcal{U}_1 = S_{\tau_2}, U_{\tau_2}$

---

**Algorithm 2** Greedy-Margin

1: **Input**: $\hat{u}, j, b, S_j, M, U_j$.
2: **Output**: $S_{j+1}, U_{j+1}$
3: $R \rightarrow$ a subset obtained by smallest margin scores $M$ examples from $U_j \setminus S_j$
4: Randomly divide $R$ into $\{\lfloor \frac{R}{b} \rfloor\}$ batches of subsets $\{(x_i)_{i=1}^b\}$.
5: $b_{\max} \leftarrow \arg\max_{\{(x_i)_{i=1}^b\} \in \{\lfloor \frac{R}{b} \rfloor\}} \hat{u}(S_j \cup (x_i)_{i=1}^b)$
6: $S_{j+1} \leftarrow S_j \cup \{b_{\max}\}$
7: $U_{j+1} \leftarrow U_j \setminus \{b_{\max}\}$

---

and then use the concatenated batch as input to $\hat{u}$ for utility prediction. We perform sequential batch selection within the acquisition stage and select the unlabeled batch with the largest predicted score.

## 4.2 THE RAMBO ALGORITHM

The essence of our two-stage utility model aligns with Shakespeare's famous line from The Tempest, "What's past is prologue." Our overarching motivation is to train an acquisition function on past utility samples that generalize well to utility samples of longer history. We initialize the utility model by collecting and training samples from offline datasets, providing an initial estimate of the *feature extractor* $\phi_0$. This initial feature extractor $\phi_0(\cdot)$ can serve as a warm start for non-adaptive batch selection in the acquisition stage. We emphasize the need for this *initialization* step as RAMBO designed for single-round acquisition.

---

**Algorithm 3** Utility-Samples-Augmentation

---

1: **Input**: $S_i$, $S_{i+1}$, $n$, $acc_i$, $acc_{i+1}$, $D_i$.
2: **Output**: $D_{i+1}$
3: **for** $j \in \text{range}(n)$ **do**
4:     Sample random a pair of $(\xi_1, \xi_2)$ from $S_i$ with equal size
5:     Compute distance between $\phi(\xi_1)$ and $\phi(S_i)$ as $d_{1,i}$ and distance between $\phi(\xi_1)$ and $\phi(S_{i+1})$ as $d_{1,i+1}$. Same Rule applies to $\xi_2$ to obtain $d_{2,i}$ and $d_{2,i+1}$.
6:     Calculate $u_1$, $u_2$ for $\xi_1$ and $\xi_2$ by Equation 4
7:     $D_i \leftarrow D_i \cup \{(\xi_1, u_1), (\xi_2, u_2)\}$
8: $D_{i+1} \leftarrow D_i$

---

#### 4.2.1 Bi-Level Optimization

To align with the growing labeled pool of AL setting, a core requirement of our utility model is the capability to *generalize to longer and unseen data* by drawing on prior utility samples. A line of research [Rajeswaran et al., 2019, Liu et al., 2019] suggests that meta-learning shall lead to fast adaptation and generalization to new tasks. One formulation of meta-learning is bi-level optimization [Maclaurin et al., 2015] where the inner objective represents the adaptation to a given task and the outer problem is the meta-training objective. Motivated by Franceschi et al. [2018], we formulate utility model training as bilevel optimization, combining gradient-based hyperparameter optimization and meta-learning in which the outer optimization problem is solved subject to the optimality of an inner optimization problem. To improve its generalization capability on samples with varied lengths, we divide the utility samples $(\xi, u(\xi))$ at iteration $i$ to training $D_{\text{tr}}$ and validation set $D_{\text{val}}$ by length, where $D_{\text{tr}}$ corresponds to utility samples with length smaller than the median and vice versa, and treat them as input dataset for *inner objective $L$* and *outer objective $E$*. We consider the bilevel optimization framework as

$$\min_{\lambda} E(w(\lambda), \lambda) \text{ s.t. } w(\lambda) = \underset{\hat{w} \in \mathbb{R}^d}{\arg\min} \mathcal{L}(\hat{w})$$

where $\lambda$ is a hyperparameter, $E$ and $\mathcal{L}$ are continuously differentiable functions, the outer objective

$$E(w(\lambda), \lambda) := \sum_{\{(S_1', u(S_1')), (S_2', u(S_2'))\} \in D_{\text{val}}} \mathcal{L}_{\text{Total}}(\hat{w})$$

and the inner objective as

$$\mathcal{L}(\hat{w}) = \sum_{\{(S_1', u(S_1')), (S_2', u(S_2'))\} \in D_{\text{tr}}} \mathcal{L}_{\text{Total}}(\hat{w}) + \Omega_{\lambda}(\hat{w})$$

where $D_{\text{tr}} = \{(\xi_1, u(\xi_1)), (\xi_2, u(\xi_2))\}_{i=1}^n$ is a set of pair of utility samples attributed to the training set, $\mathcal{L}_{\text{Total}}(\cdot)$ is the loss function specified in Definition 4, and $\Omega_{\lambda}$ is a regularizer parametrized by $\lambda$. The outer objective is the proxy of

the generalization error of $\hat{u}(\cdot)$, given by the average loss on $D_{\text{val}}$.

The inner optimization is aimed at *utility model optimization*, i.e., finding the best parameters that minimize the loss on smaller length training samples $D_{\text{tr}}$. Conversely, the outer optimization targets to *generalize the model to longer-length utility samples $D_{\text{val}}$*, which seeks the optimal regularizer parameterized by $\lambda$. With bilevel formulation, RAMBO shows better and more stable performance when performing unlabeled data selection on CIFAR10 with labeling budge 5000 (as suggested by Table 1). Table 1 shows the average performance of models with bilevel training used in optimization, which mostly outperforms the rest of counterparts without bilevel training, illustrating the enhanced generalizability across various model architectures and training algorithms.

#### 4.2.2 Interpolation-Based Utility Samples

In contrast to thousands or even millions of training samples for datamodels framework Ilyas et al. [2022], Engstrom et al. [2024], the scarcity of utility samples poses challenges to the efficacy of our utility model training. We resort to the consistency regularization techniques from semi-supervised learning to augment artificial $(\xi, u(\xi))$. Inspired by Parvaneh et al. [2022], the latent space of the classifier's feature extractor shall contain valuable representations that can be interpolated within labeled instances. The empirical success suggests a change in perspective—rather than twisting the classifier, we leverage the shared representations in $\hat{u}$ throughout the progress of optimization. In particular, we adopt the *interpolation consistency regularization* strategy [Verma et al., 2022] (Definition 5). The pseudo code for utility samples augmentation is outlined in Algorithm 3.

**Definition 5** (Utility Value Interpolation). *Denote the validation accuracy at iteration $i$ as $acc_i$. For a given utility sample $\xi_1$, let $d_{1,i}$ be its distance [1] with the previous labeled pool $S_i$ and $d_{1,i+1}$ the distance with the current labeled pool $S_{i+1}$. The augmented utility value $u_1$ for $\xi_1$ yields as*

$$u_1 = \alpha \cdot u_i + (1 - \alpha) \cdot u_{i+1} \tag{4}$$

*with $\alpha := \frac{d_{1,i+1}}{d_{1,i+1} + d_{1,i}}$.*

## 5 EXPERIMENTAL RESULTS

### 5.1 EXPERIMENTAL SETUP

We evaluate the performance of RAMBO on four active learning benchmarks in image classification: MNIST [LeCun et al., 1998], FashionMNIST [Xiao et al., 2017], CIFAR10 [Krizhevsky et al., 2009], SVHN[Netzer et al., 2011].

---

[1]The OT distance is computed by utilizing utility model latent space representation.

To facilitate a thorough comparison against the baselines, we evaluated them across various acquisition stage budget $B$ as $\{500, 700, 900, 1000\}$ for MNIST and FashionMNIST with $k = 200$, $\{5000, 7000, 9000, 10000\}$ for CIFAR10 and SVHN with $k = 2500$. We focus on the accuracy of the validation set as the key performance metric, with the validation set size fixed at 1000 for all datasets. We ran each experiment ten times and reported average and standard error across all experiments.

We consider two network architectures: For MNIST and FashionMNIST, we utilized a neural network structure similar to LeNet [LeCun et al., 1998], as suggested by Beck et al. [2021], and for CIFAR10 and SVHN, we employed ResNet-18 [He et al., 2016]. We defer the details of utility model architecture and the choice of classifiers to the Appendix A.

We fit all classifiers using cross-entropy loss with the Adam optimizer until training accuracy exceeds 99% with maximum 100 epochs and learning rate 0.001. No learning rate schedulers or data augmentations are used. [2]

## 5.2 BASELINES

While numerous AL methods has been proposed for specific tasks such as object detection [Yuan et al., 2021], semantic segmentation [Kim et al., 2021] and instance segmentation [Chaplot et al., 2021], these algorithms rely on heuristics acquisition functions and are not suitable to the single-round setting considered in this work. Hence, we mainly consider the state-of-the-art learning-based AL baselines designed for the single-round AL setting:

**DULO** [Wang et al., 2023]: A learning based approach curated for one round AL setting by selecting a subset with size $B$ instances from $\mathcal{U}_0$ which maximize a learned utility function. DOLO relies on a regression-based surrogate utility function, and employs a stochastic block-wise greedy selection strategy for batch acquisition. In contrast, our algorithm utilizes a RankNet with a multi-task training loss for training the acquisition function, and uses the greedy-margin subroutine for data acquisition.

**LLAL** [Yoo and Kweon, 2019]: A learning based approach estimating the errors of the predictions (loss) made by the classifier and select $B$ unlabeled instances with top predicted losses. [3]

We also include a RANDOM strategy as baseline, which selects $B$ samples uniformly at random. For additional comparisons against a collection of non-task-aware, heuristic-

based AL baselines, please refer to the supplemental results in Appendix B.1.

## 5.3 MAIN RESULTS

In Figure 2, RAMBO outperforms most of the baselines across multiple architectures and various labeling budgets for the acquisition stage. For easy datasets like Fashion-MNIST and MNIST, RAMBO shall learn a good shared representation for effective utility value interpolation and can easily beat all the baselines oblivious to different labeling budgets which suggests RAMBO is a good choice regardless of labeling budget. Even though MNIST and FashionMNIST are easy to learn, but due to limited labeled pool, LLAL [Yoo and Kweon, 2019] fails to learn a good loss prediction module or *uncertainty* estimate, and thus RAMBO has a substantial gain compared to it. However, RAMBO performs interpolation techniques to augment utility samples within a limited labeled pool and generalize to predictions of a longer history of labeled data, leading to a learning-based acquisition function amenable to the growing labeled pool. For more challenging datasets, such as CIFAR10 and SVHN, when the model fails to have good architecture priors due to a limited labeled pool, RAMBO outperforms DULO Wang et al. [2023] in large gain compared to easy datasets. We conjecture it's because ranking is generally easier for model to learn compared to regression, especially under complex dataset (Figure 1).

## 5.4 ABLATION STUDY

We perform an ablation study on the size of Pretraining set, the design choices of each submodule as bilevel training, OT distance and RankNet as well as hyperparameter for OT Distance Loss (Definition 4). We use CIFAR10 as an example dataset, and defer our results on the remaining datasets to the Appendix B.

**Size of pretraining budget** $k$ Naturally, we want to examine the effect of size of pretraining set for determining how the scale of initial labeled pool impacts overall single round selection performance. Figure 2e shows across different seed set size for pretraining stage, RAMBO outperforms all other baselines.

**Bi-level training, OT Distance and RankNet** Next, we shift to study the intertwined effects of three design choices. Table 1 shows the combined efficacy of bilevel training, OT distance, and RankNet, offering insights into the synergy of these three foundational modules. The cross mark for RankNet means regression based acquisition function and the loss is designed as MSE between predicted utility vs. true utility value. One thing to note is that if the performance of regression based acquisition function without bi-level training and OT distance is similar to random,

---

[2]Baselines use implementations from open-source AL toolkit DISTIL Team [2023]. All models are trained in PyTorch [Paszke et al., 2017].

[3]Yoo and Kweon [2019] design loss prediction module using middle layers of ResNet18. For FashionMNIST and MNIST, we extract middle layers of Beck et al. [2021]'s neural networks.

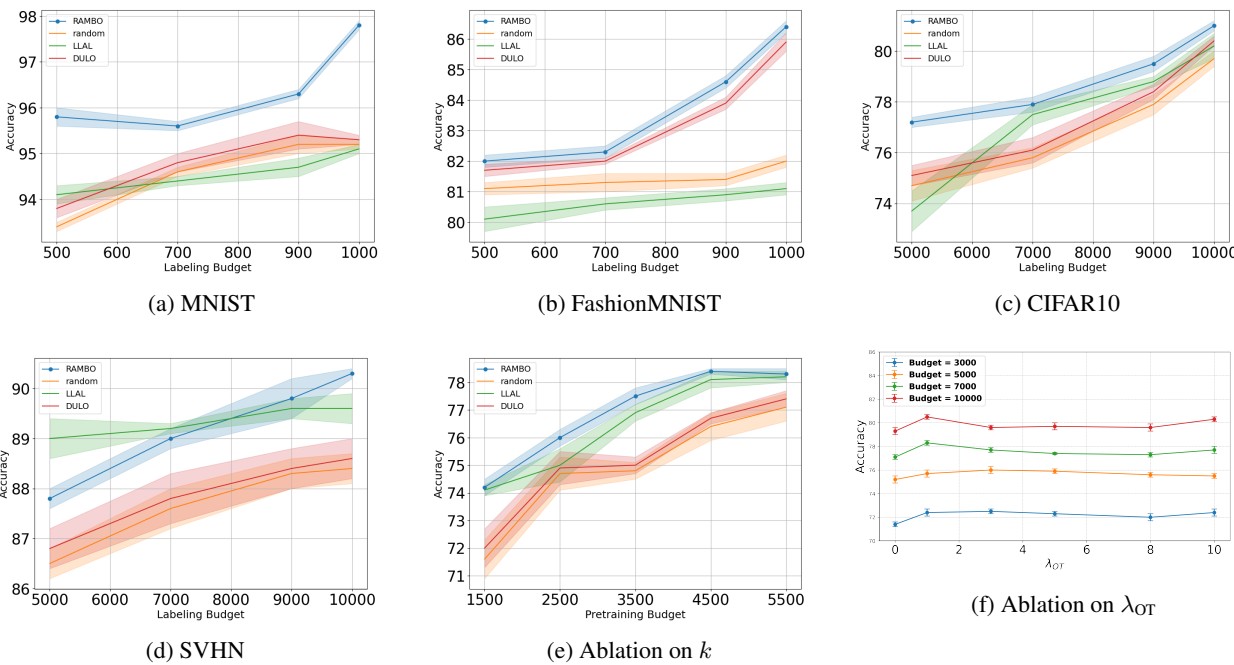

(a) MNIST      (b) FashionMNIST      (c) CIFAR10

(d) SVHN      (e) Ablation on $k$      (f) Ablation on $\lambda_{\mathrm{OT}}$

Figure 2: Experimental results. **(a-d)** Active learning validation performance. **(e)** Active Learning validation performance with the acquisition stage budget $B = 5000$ for CIFAR10 across various choices of pretraining budget $k$. **(f)** Different choices of $\lambda_{\mathrm{OT}}$ for pretraining set size $k = 2500$ on CIFAR10. Results are given in %.

which corroborates our intuition about ranking instead of regressing validation accuracy on labeled samples.

Table 1: Ablation study on three submodules with pretraining set $k = 3500$ and acquisition budget $B = 5000$. The last row corresponds to the random baseline.

| Bilevel | Optimal Transport | RankNet | Accuracy |
|---------|-------------------|---------|----------|
| ✓ | ✓ | ✓ | **77.3 ± 0.2** |
| ✓ | ✓ | ✗ | 76.1 ± 0.3 |
| ✓ | ✗ | ✓ | 76.2 ± 0.4 |
| ✓ | ✗ | ✗ | 70.5 ± 0.3 |
| ✗ | ✓ | ✓ | 75.5 ± 0.3 |
| ✗ | ✓ | ✗ | 75.5 ± 0.3 |
| ✗ | ✗ | ✓ | 76.0 ± 0.8 |
| ✗ | ✗ | ✗ | 74.6 ± 0.7 |
| - | - | - | 74.7 ± 0.3 |

**Hyperparameter Tuning for OT distance** By definition, $\mathcal{L}_{\mathrm{Total}} = \mathcal{L}_{\mathrm{Rank}_{12}} + \lambda_{\mathrm{OT}} \cdot \mathcal{L}_{\mathrm{OT}}$ (Definition 4). One can change the scale of $\lambda_{\mathrm{OT}}$ for utility model training in pretraining. We study the effect of hyperparameter $\lambda_{\mathrm{OT}}$ in final model performance on validation set. We highlight the importance of incorporating OT distance into the loss structure which makes $\hat{u}$ insensitive to the scale of $\lambda_{\mathrm{OT}}$. When $\lambda_{\mathrm{OT}} > 0$, the overall validation accuracy is larger than $\lambda_{\mathrm{OT}} = 0$. The choice of $\lambda_{\mathrm{OT}}$ is specific to dataset and batch setting and we present one setting of $\lambda_{\mathrm{OT}}$ with varied Labeling Budget for acquisition stage in Figure 2f. We also provide additional

results on more fine-grained orders of magnitude of $\lambda_{OT}$ in Appendix B.6.

## 5.5 ROBUSTNESS ANALYSIS

**Validation set size vs. validation accuracy** One potential concern of surrogate model training is the consistency and robustness of *utility* across different subset sizes. In real-world applications, it is crucial to adapt to scenarios with varying data availability, ranging from scarce data, resulting in small validation sets [Hacohen et al., 2022], to situations with ample labeled examples, leading to larger validation sets [Citovsky et al., 2021]. To account for the variations in validation set size, we conduct experiments to measure validation set accuracy across various sizes. Using CIFAR10 as a benchmark dataset, we evaluated 100 randomly collected utility samples, assessing utility values across validation set sizes of 200, 400, 600, 800, 1000. As illustrated in Figure 3a, the average validation set accuracy remains consistent regardless of the validation set size, with the standard error decreasing as the size of the validation set increases. We have also conducted sensitivity analysis of validation accuracy w.r.t the size of validation set size for MNIST, FashionMNIST and SVHN in Appendix B.2.

**Noisy oracles** The quality of labels provided by an oracle can vary depending on the expertise of human annotators. For example, labels from medical images annotated by ex-

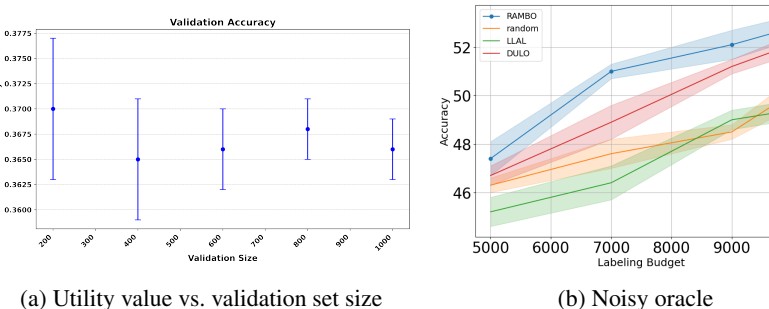

| (a) Utility value vs. validation set size | (b) Noisy oracle | (c) Class Imbalance |

Figure 3: Robustness analysis.

perts are likely to be more accurate compared to crowd-sourced data from non-experts. To examine the robustness of RAMBO, we investigate the impact of a noisy oracle, which non-adversarially generated erroneous labels for certain classes. We randomly changed the groundtruth labels for 20% of the data to reflect incorrect labeling. Figure 3b denotes RAMBO outperforms the rest of three baselines by a large margin even though all of four methods are greatly affected by noisy labels with performances dropped by 30%.

**Class-Imbalance** The issue of class imbalance, where some classes are underrepresented compared to others, can significantly affect the performance of active learning algorithms. To investigate the robustness of RAMBO in such scenarios, we follow class-imbalanced settings similar to Killamsetty et al. [2021] and artificially generate class-imbalance for the above dataset by removing 20% of the instances from 30% of total classes available. Figure 3c illustrates that RAMBO exhibits a much greater advantage over other baselines.

## 6 CONCLUSION

We have demonstrated existing state-of-the-art methods can be suboptimal in single round selection. We show that under certain budget for pretraining, RAMBO shall achieve better generalization performance compared to other active learning algorithms, and that most of validation accuracy improvement is realized by our two-stage algorithm. Finally, we illustrate how behaviors of all algorithms change with variation of pretraining and single round acquisition budget across multiple datasets and architectures. One potential direction for future work could be to determine an optimal budget allocation for both the pretraining and acquisition stages, as well as the extension of RAMBO to the few-rounds setting.

### Acknowledgements

We are thankful to Feiyang Kang and Jiachen T. Wang for providing useful discussion and helpful feedback on the

paper. YC and ZD acknowledge support from the National Science Foundation under Grant No. NSF IIS-2313131 and NSF FMRG-2037026. RJ and the ReDS lab acknowledge support through grants from the Amazon-Virginia Tech Initiative for Efficient and Robust Machine Learning, the National Science Foundation under Grant No. IIS-2312794, NSF IIS-2313130, NSF OAC-2239622, and the Commonwealth Cyber Initiative.

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

# A   UTILITY MODEL ARCHITECTURE AND CHOICE OF CLASSIFIERS

Here, we describe the acquisition network (Utility Model) discussed in Section 5.

## A.1   MNIST AND FASHIONMNIST

Our architecture performs the following operations on pairs of subsets of images (Utility Samples) with equal size. We use below networks as feature extractor for pairs of raw embeddings of images. For each one within the pair:

1. 2-D Convolution on set of images.

2. 2-D Max Pool on output of (1).

3. ReLU on output of (2).

4. 2-D DropOut on output of (3).

5. 2-D Max Pool on output of (4).

6. ReLU on output of (5).

7. Fully-Connected Layer on Output of (6).

8. ReLU on output of (7).

9. 2-D DropOut on output of (8).

10. Fully-Connected Layer on output of (9).

11. ReLU on output of (10).

## A.2 CIFAR10 AND SVHN

We use pretrained ResNet-18 on ImageNet as feature extractor and perform the following operations on pairs of subsets of extracted features for each image. For each one within the pair:

1. Fully Connected Layer on set of feature embeddings.

2. ReLU on output of (1).

3. Fully Connected Layer on output of (2).

## A.3 MUTITASK SET-BASED NEURAL NETWORKS WITH RANKNET

After average pooling of output of (11) for MNIST and FashionMNIST in Section A.1 and output of (3) for CIFAR10 and SVHN in Section A.2,

for each one within the pair, we perform the following operations:

1. Fully Connected Layer on extracted features

2. ReLU on output of (1).

3. Fully Connected Layer on output of (2).

Denote the output of (3) as $\phi_1$ and $\phi_2$ for both the first one and second one in each pair of utility sample.

For the prediction of probability score that which subset has larger utility value in the pair, we apply RankNet on $\phi_1$ and $\phi_2$ for pair comparison. The output score predicted by RankNet is the final probability score that we shall use to determine whether the first set has larger utility value than the second.

For the interpolation of utility value, we use $\phi_1$ and $\phi_2$ as embedding. For computing the distance between above two embeddings, we resort to the Euclidean distance.

For the prediction of optimal transport distance, we use MLP projection head for $\phi_1$ and $\phi_2$:

1. Fully Connected Layer on $\phi_1$ and $\phi_2$

2. ReLU on outputs of (1)

3. Fully-Connected Layer on outputs of (2).

We use the outputs of (3) as a supervision signal in designing the loss function for the utility model (see Definition 3 in Section 4).

We choose $\lambda_1, \lambda_2$ to be 0.5 and $\lambda_3$ to be 1.

## A.4 CHOICE OF CLASSIFIERS

The reason why we do not use ResNet-18 for MNIST type datasets is that ResNet-18 might be an overkill for MNIST and FashionMNIST as MNIST is a relatively simple dataset

consisting of grayscale images of handwritten digits with a resolution of 28x28 pixels. ResNet18 is a complex architecture designed for much more challenging image recognition tasks. Moreover, due to its depth and complexity, ResNet18 has a lot more parameters compared to simpler models. Training such a large model on a 700 data points for MNIST could lead to overfitting with poor generalization. In fact, with 700 labeled MNIST data points, the ResNet-18 structure only achieves, on average, 78 % validation set accuracy.

# B SUPPLEMENTAL EXPERIMENTAL RESULTS

## B.1 ADDITIONAL AL BASELINES

We also consider non-task-aware and representative deep AL baselines. For all experiments, we include a classical Margin Sampling algorithm, two recent active learning algorithms, BADGE and CoreSet, one learning-based algorithm, GLISTER, and random selection Random.

**Margin Sampling** [Roth and Small, 2006]: Selects $B$ examples from $\mathcal{U}_0$ with the smallest difference between the first and second most probable classes predicted by $f$.

**BADGE** [Ash et al., 2019]: A hyperparameter-free approach that trades between diversity and uncertainty using k-means++ in hallucinated gradient space.

**CoreSet** [Sener and Savarese, 2017]: A diversity-based approach using greedy approximation to the k-center problem on representations from the current classifier's penultimate layer.

**GLISTER** [Killamsetty et al., 2021]: A learning-based approach selecting $B$ instances from $\mathcal{U}_0$ that would maximize the log-likelihood on held-out validation set $\mathcal{S}_{val}$ by converting it as a mixed discrete-continuous bilevel optimization. We adopt the GLISTER-ONLINE version as an approximation for the inner optimization problem by taking a single gradient step update.

## B.2 VALIDATION SET SIZE VS. VALIDATION ACCURACY

In addition to Fig3a, we provide additional results on validation set accuracy w.r.t the size of the validation set (averaged across ten trials) in Table 2. These results demonstrate the robustness of validation accuracy as a consistent measure of utility value, with the standard error inside the parentheses generally decreasing as the validation set size increases. Even at smaller validation set sizes of 50 or 100—significantly less than the pretraining set size $\mathcal{S}_0 = k$—the accuracy measures are comparable to those observed with larger sizes, such as 800 or 1000.

| Dataset\validation size | 50 | 100 | 200 | 400 | 600 | 800 | 1000 |
|---|---|---|---|---|---|---|---|
| **SVHN** | 0.132(0.018) | 0.132(0.012) | 0.133(0.011) | 0.130(0.010) | 0.136(0.009) | 0.130(0.009) | 0.130(0.008) |
| **MNIST** | 0.758(0.007) | 0.757(0.005) | 0.763(0.001) | 0.764(0.003) | 0.760(0.009) | 0.770(0.009) | 0.764(0.008) |
| **FashionMNIST** | 0.699(0.023) | 0.714(0.015) | 0.707(0.011) | 0.712(0.009) | 0.712(0.009) | 0.707(0.009) | 0.712(0.009) |
| **CIFAR10** | 0.358(0.020) | 0.372(0.016) | 0.370(0.007) | 0.365(0.008) | 0.366(0.005) | 0.368(0.003) | 0.366(0.003) |

Table 2: Validation performance across different datasets and validation sizes.

## B.3 ADDITIONAL RESULTS ON CLEAN DATA, NOISY ORACLES, AND CLASS-IMBALANCE SETTINGS

We follow the settings and present additional results, including both the learning-based, one-round AL baselines discussed in the main paper and the deep AL baselines described in Appendix B.1. In particular, we focus on all scenarios, including the default clean data setting, the noisy oracle setting, and the class-imbalance setting for all benchmarks (MNIST, FashionMNIST, CIFAR10 and SVHN) in Figure 4, Figure 5 and Figure 6.

RAMBO outperforms the rest of the baselines in most scenarios. Admittedly, for SVHN, LLAL [Yoo and Kweon, 2019] outperforms the rest of the baselines (including RAMBO) by a large margin. Indeed, SVHN is an easy dataset with a large initial pool of $k$ and a labeling budget of $B$. Given $k = 2500$, as mentioned in Hacohen et al. [2022], uncertainty plays a much more significant role than diversity when the labeling budget and initial labeled set are both important. Training a classifier with a reasonably accurate uncertainty estimate is feasible. Therefore, the specific design choice of LLAL [Yoo and Kweon, 2019] to estimate the cross-entropy loss between pairs of unlabeled instances, another measure of uncertainty but with *groundtruth* labels information incorporated in the loss prediction module, shall have superior empirical results in SVHN. Yet, we emphasize that LLAL is non-robust across different datasets. For instance, LLAL has mediocre performance in Figure 3c (a) and (b), much less than RAMBO. One similar argument could be FashionMNIST and MNIST are easy datasets and thus Hacohen et al. [2022] suggest that the acquisition function should focus on typical, easy and representative points.

Moreover, compared to DULO, RAMBO requires much fewer samples for training. Prior works either involve training millions of datamodels [Engstrom et al., 2024] or collecting thousands of samples [Wang et al., 2023], which would require considerable time before the deployment or acquisition stage. In contrast, ours requires only hundreds of utility samples to achieve a fair amount of accuracy improvement. This efficiency is achieved by imposing a strong regularization signal through the OT distance loss, and by reducing the regression task to a ranking problem.

## B.4 SIZE OF PRETRAINING SET

In the main paper, we have focused our evaluation on CIFAR-10. Here, we provide experiments to show the effectiveness of RAMBO on diverse datasets such as MNIST, FashionMNIST, and SVHN for single-round unlabeled data selection. We construct all the pretraining sets by random sampling from the whole training set of each dataset.

Figure 7 illustrates the impact of the size of the pretraining set on final validation set accuracy. One shall see RAMBO outperforms the rest of the baselines with most of the pretraining splits. The only performance degradation case of RAMBO could be SVHN where $k = 5500$ and $B = 5000$. One possibility could be $k = 5500$ is suffice for BADGE to learn an accurate-enough gradient embedding space for single round selection. Therefore, BADGE could beat RAMBO when $k = 5500$ and $B = 5000$ for SVHN as the pretraining set is sufficiently large compared to the acquisition budget. Another interesting observation is that GLISTER often performs worse than most baselines for three datasets when the pretraining set has an extremely low budget, as $k = 100$ for FashionMNIST/MNIST and $k = 1500$ for SVHN. A plausible reason could be that a limited pretraining budget, combined with a substantial acquisition budget, might exacerbate the bias brought about by the single-step gradient approximation during the inner-level optimization phase, particularly when trying to maximize the log-likelihood of the training set.

## B.5 BILEVEL TRAINING, OT DISTANCE AND RANKNET

For simplicity, the "✓" for optimal transport denotes $\lambda_{OT} = 1$ and the "×" for RankNet represents regression-based utility model as stated in the main paper. For ablating other network components, we still keep the same feature extractor explained in Section A.1 for MNIST and FashionMNIST and Section A.2 for CIFAR10 and SVHN. For the regression style acquisition function, we impose MLP head on the shared representation space $\phi$ for predicting validation accuracy with $\hat{u} = g(\phi) = W^{(2)}(\sigma(W^{(1)}))$ where $\sigma$ is a RELU activation function, very much similar to the description of predicting OT distance in Section A.3. For OT distance regularization, we adopt the same MLP projection head architecture described in Section A.3.

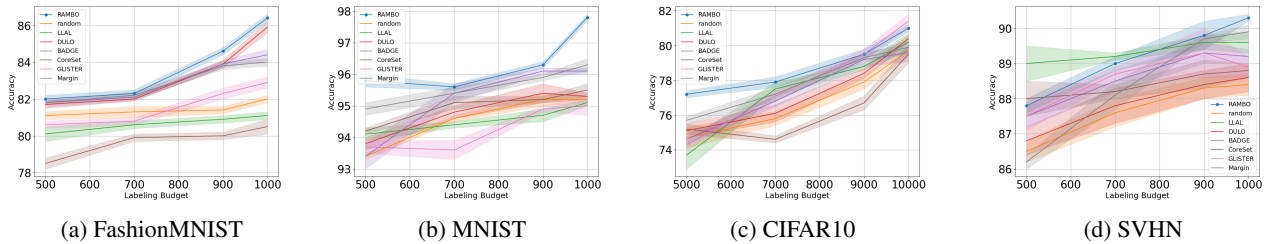

Figure 4: Clean data setting: Active learning validation performance with $B = 500$ for FashionMNIST and MNIST and $B = 5000$ for CIFAR10 and SVHN. Results are given in %. The shaded area denotes standard error.

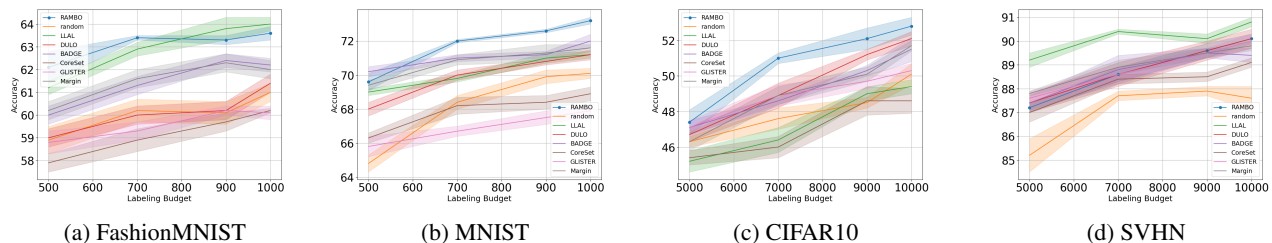

Figure 5: Noisy oracles setting: Active learning validation performance with $B = 500$ for FashionMNIST and MNIST and $B = 5000$ for CIFAR10 and SVHN. Results are given in %. The shaded area denotes standard error.

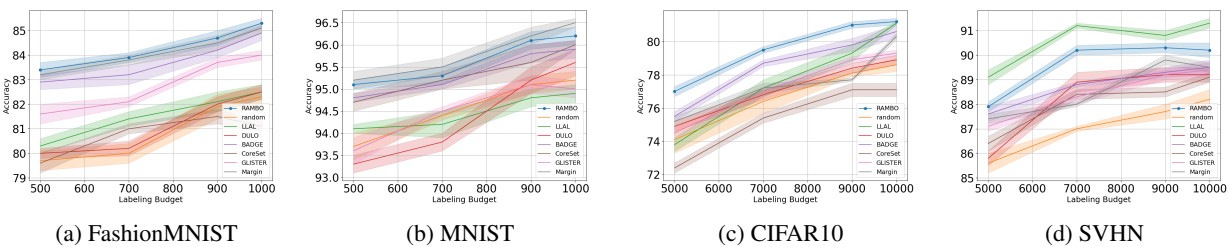

Figure 6: Class-imbalance setting: Active learning validation performance with $B = 500$ for FashionMNIST and MNIST and $B = 5000$ for CIFAR10 and SVHN. Results are given in %. The shaded area denotes standard error.

Table 3: Ablation study on three submodules with pretraining budget $k = 200$ and acquisition budget $B = 500$ for FashionMNIST. The last row corresponds to the random baseline.

| Bilevel | Optimal Transport | RankNet | Accuracy |
|---------|-------------------|---------|----------|
| ✓ | ✓ | ✓ | **83.1 ± 0.1** |
| ✓ | ✓ | ✗ | 81.9 ± 0.2 |
| ✓ | ✗ | ✓ | 81.2 ± 0.4 |
| ✓ | ✗ | ✗ | 81.8 ± 0.2 |
| ✗ | ✓ | ✓ | 81.0 ± 0.3 |
| ✗ | ✓ | ✗ | 81.7 ± 0.2 |
| ✗ | ✗ | ✓ | 80.9 ± 0.3 |
| ✗ | ✗ | ✗ | 81.6 ± 0.1 |
| - | - | - | 81.2 ± 0.2 |

Table 4: Ablation study on three submodules with $k = 200$ and $B = 500$ for MNIST. The last row corresponds to the random baseline.

| Bilevel | Optimal Transport | RankNet | Accuracy |
|---------|-------------------|---------|----------|
| ✓ | ✓ | ✓ | **95.3 ± 0.2** |
| ✓ | ✓ | ✗ | 94.9 ± 0.2 |
| ✓ | ✗ | ✓ | 95.0 ± 0.1 |
| ✓ | ✗ | ✗ | 94.8 ± 0.2 |
| ✗ | ✓ | ✓ | 94.6 ± 0.1 |
| ✗ | ✓ | ✗ | 94.9 ± 0.1 |
| ✗ | ✗ | ✓ | 95.0 ± 0.2 |
| ✗ | ✗ | ✗ | 94.8 ± 0.2 |
| - | - | - | 93.4 ± 0.1 |

To prove the efficacy of synergizing three seemingly irrelevant submodules together, we provide ablation study of

three submodules for the rest of three datasets. Table 3, 4 and 5 show the impact of turning off each submodule on the final validation set accuracy for FashionMNIST, MNIST

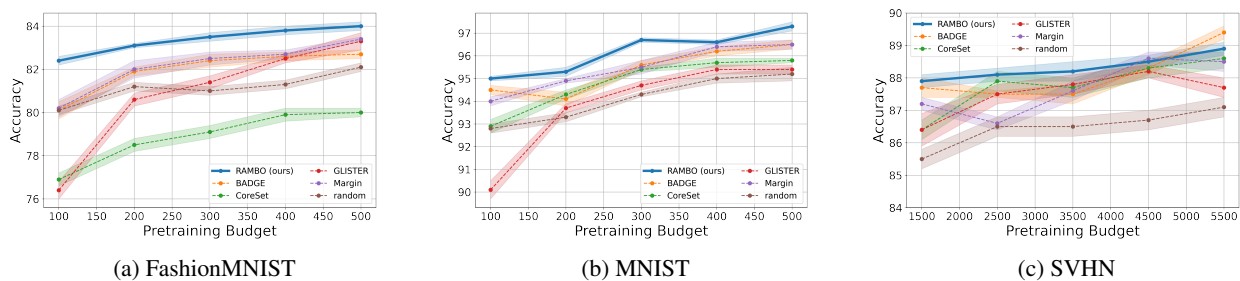

(a) FashionMNIST       (b) MNIST       (c) SVHN

Figure 7: Experimental results. **(a-c)** Active learning validation performance with $B = 500$ for FashionMNIST and MNIST and $B = 5000$ for SVHN. Results are given in %. Shaded area denotes standard error.

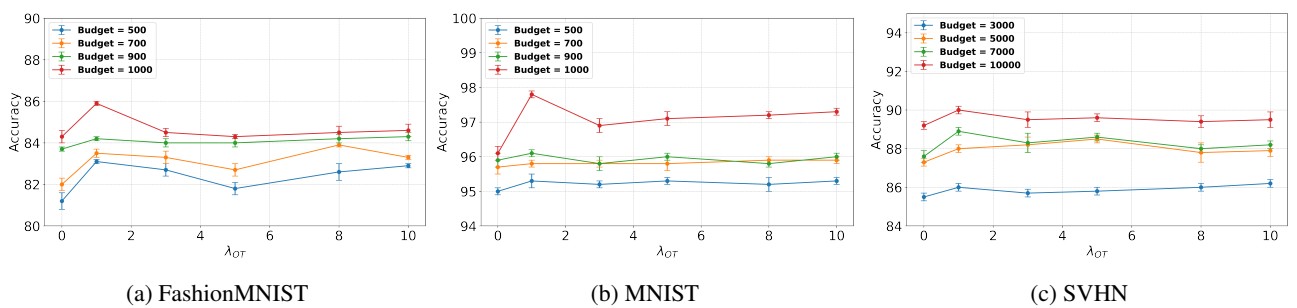

(a) FashionMNIST       (b) MNIST       (c) SVHN

Figure 8: Ablation on $\lambda_{OT}$ across different acquisition budget. **(a-b)** $k = 200$; **(c)** $k = 2500$

.

Table 5: Ablation study on three submodules with $k = 3500$ and $B = 5000$ for SVHN. The last row corresponds to the random baseline.

| Bilevel | Optimal Transport | RankNet | Accuracy |
|:---:|:---:|:---:|:---:|
| ✓ | ✓ | ✓ | **88.1 ± 0.3** |
| ✓ | ✓ | ✗ | 86.7 ± 0.2 |
| ✓ | ✗ | ✓ | 87.8 ± 0.3 |
| ✓ | ✗ | ✗ | 86.5 ± 0.3 |
| ✗ | ✓ | ✓ | 87.8 ± 0.2 |
| ✗ | ✓ | ✗ | 86.3 ± 0.1 |
| ✗ | ✗ | ✓ | 87.5 ± 0.1 |
| ✗ | ✗ | ✗ | 86.1 ± 0.2 |
| - | - | - | 86.5 ± 0.3 |

and SVHN respectively.

While the main premise of combining three submodules to improve validation set performance, it is natural to evaluate the significance of each submodule plays the role in utility model training. For FashionMNIST, one might see bilevel training plays a signicant role in obtaining good validation set accuracy as the top three combinations of submodules all have bilevel training turned on (Table 3). For MNIST, the gain of validation set accuracy for each design choice shall be subtle to differentiate under various settings as deterministic CNNs can easily achieve 96% accuracy under simpler acquisition heuristics ,i.e., least confidence or max entropy

[Gal et al., 2017]. Still, we shall see the marginal improvement with all three submodules turned on (Table 4) and the top three combinations of submodules all have RankNet turned on which validates the necessity of pairwise ranking in the design choice of our utility model. For complex datasets as SVHN, RankNet plays a crucial role in improving classification performance as expected, demonstrated by the top three scores of accuracy all have RankNet turned on (Table 5).

## B.6   HYPERPARAMETER TUNING FOR OT DISTANCE

Figure 2(f) in Section 5 illustrates the benefits of incorporating optimal transport distance into the loss structure of our utility model. Figure 8 shall serve as a complement to uncover the usefulness of optimal transport distance, regardless of the scale of $\lambda_{OT}$, for various datasets of interest. Regardless of datasets and classification networks architecture, the incorporation of optimal transport distance finds utility in reducing generalization error, measured by the increase of validation set accuracy. Even though $\lambda_{OT}$ can be a hard hyperparameter for fine-tuning, either Figure 2(f) and Figure 8 suggest final validation set accuracy for $\lambda_{OT} \neq 0$ is higher than its counterpart for $\lambda_{OT} = 0$.

For additional results on CIFAR10 with $\lambda_{OT}$ spanning more orders of magnitude, please see Table 6.

| Labeling Budget \\$\lambda_{OT}$ | 0 | 10e-2 | 10e-1 | 1 | 10 | 100 |
|---|---|---|---|---|---|---|
| **3000** | 0.714(0.002) | 0.718(0.006) | 0.728(0.003) | 0.721(0.002) | 0.724(0.003) | 0.718(0.005) |
| **5000** | 0.752(0.003) | 0.761(0.004) | 0.760(0.003) | 0.772(0.002) | 0.755(0.003) | 0.757(0.004) |
| **7000** | 0.771(0.003) | 0.774(0.004) | 0.779(0.005) | 0.778(0.003) | 0.779(0.003) | 0.788(0.009) |
| **10000** | 0.793(0.003) | 0.800(0.003) | 0.801(0.004) | 0.809(0.002) | 0.803(0.002) | 0.816(0.005) |

Table 6: Labeling budget vs. $\lambda_{OT}$ performance. Values in parentheses indicate standard error.

### B.7 RUNTIME ANALYSIS

All models are trained using NVIDIA A40 GPU with 48GB. To increase the running speed of our experiments, we use data parallelism on multiple GPUs in implementations. The time recorded below is for Pytorch training with 2 GPUs. As stated in main text, all the experiments are repeated for 10 trials to reduce the training stochasticity. We fix $k = 2500$ and $B = 5000$ for CIFAR10 and SVHN with $n = 30$ utility samples collected per batch with $\tau_1 = 2$, $b = 1000$ and $k_1 = 500$ for pretraining stage with each batch trained for 20 epochs. For CIFAR10, we collect 500 pairs of utility samples for $\hat{u}$ offline training with roughly 3 hours and 20 minutes. Then, the total training time for both pretraining and acquisition stage is 1 hour and 20 minutes with pretraining stage 50 minutes and acquisition stage 30 minutes. For SVHN, we collect 500 pairs of utility samples for offline training with roughly 1 hour and 40 minutes. Then, the total training time for both pretraining and acquisition stage is roughly 1 hour with pretraining stage 29 minutes and acquisition stage 34 minutes.

We fix $k = 200$ and $B = 500$ for MNIST and FashionMNIST with $n = 50$ utility samples collected per batch with $\tau_1 = 3$, $b = 50$ and $k_1 = 50$ for pretraining stage with each batch trained for 20 epochs. For MNIST, we first randomly collect 500 pairs of utility samples and the total training time for utility model $\hat{u}$ for offline training is 50 minutes. Then, the total training time for both pretraining and acquisition stage is 59 minutes with pretraining stage 39 minutes and acquisition stage 20 minutes. For FashionMNIST, the offline training for utility model $\hat{u}$ is 50 minutes for 500 pairs of utility samples. The total training time for both pretraining and acquisition stage is 50 minutes with pretraining stage 32 minutes and acquisition stage 18 minutes.

For learning-based acquisition function LLAL [Yoo and Kweon, 2019], the training time for loss prediction module for CIFAR10 with $k = 2500$ and $B = 5000$ is 20 minutes and acquisition time is 50 minutes and SVHN with $k = 2500$ and $B = 5000$ is 5 minutes and acquisition time is 30 minutes. For Margin, GLISTER, random, BADGE and CoreSet applied on CIFAR10 and SVHN, the training time for pretraining set is roughly 5 minutes and acquisition stage is 14 minutes, 20 minutes, 10 minutes, 20 minutes and 10 minutes. For those four methods applied on FashionMNIST and MNIST, the training time for pretraining and acquisition stage are negligible.