# OpenReview forum: "Learning to Rank for Active Learning via Multi-Task Bilevel Optimization"
_auai.org/UAI/2024/Conference — UAI 2024 poster_

### Official Review · Reviewer_nEAz · 2024-03-19

**Q2-1 Originality-Novelty:** 2
**Q2-2 Correctness-Technical Quality:** 3
**Q2-5 Clarity Of Writing:** 1

**Q1 Summary And Contributions:**

This paper presents a method for active learning with a single round of annotation.  The method comprises two steps:

1.	Pre-train a surrogate model for predicting the utility of an input set of samples. Here, utility is defined as the validation accuracy achieved by employing the set for training. The training set is derived from the labeled samples.
2.	Iteratively employ the utility model to select a batch of samples from the M smallest-margin unlabeled samples (not yet selected). This selection process is conducted greedily, prioritizing the batch with the highest estimated utility.

The paper provides comprehensive analysis results, which encompass a comparison of the algorithm against three other baselines (one of which is “random selection”) across various conditions, as well as an ablation study.

**Q2-3 Extent To Which Claims Are Supported By Evidence:**

3: Good: the main claims are supported by convincing evidence (in the form of adequate experimental evaluation, proofs, (pseudo-)code, references, assumptions).

**Q2-4 Reproducibility:**

1: Poor: key details (e.g. proof sketches, experimental setup) are incomplete/unclear, or key resources (e.g. proofs, code, data) are unavailable.

**Q3 Main Strengths:**

1.	The paper addresses an important problem with practical relevance.
2.	The method's core concept is intriguing and effectively conveyed throughout the paper.
3.	The algorithm's analysis is comprehensive, yielding compelling evidence regarding the method's effectiveness and robustness.

**Q4 Main Weakness:**

1.	The algorithm lacks a truly novel idea, instead presenting a combination of established techniques, albeit in a novel arrangement, to the best of my knowledge.
2.	Certain technical details in both the algorithm and the analysis are inadequately explained. Please refer to the detailed comments below for clarification.

**Q5 Detailed Comments To The Authors:**

1.	The rationale behind utilizing the optimal transport (OT) distance as a supervision signal lacks complete clarity (page 4): “The celebrated success of OT distance in predicting validation set accuracy …”. Specifically, it's unclear how OT can be leveraged to predict validation set accuracy. Is it because a smaller OT distance between two samples implies that their induced validation set accuracy will be similar as well?
2.	Figure 1: Please adjust the figure to clarify that both input set 1 and input set 2 are processed by the utility function $u$.
3.	Algorithm 1:  Please provide definitions for each argument in the input to ensure the algorithm is self-contained.
4.	Algorithm 1, Line 5: Define $k=|\mathcal{S}_0|$ to avoid the need to search for definitions throughout the paper.
5.	Algorithm 1, Line 14:   Clarify the rationale for training $\hat{u}_i$ for $i < \tau_1$.  It appears that$\hat{u}_i$ is only utilized for $i= \tau_1$  in line 16, so the necessity of training earlier instances should be justified.
6.	Algorithm 3:  Please include definitions for the input arguments to ensure clarity.
7.	Algorithm 3, Line 2: Correct the output notation to $D_{i+1}$
8.	Algorithm 3, Line 3: Ensure clarity by avoiding the use of $i$ as the loop variable, as it has already been utilized to annotate $S_i, S_{i+1}$
9.	Algorithm 3, Line 4: Describe how $\xi_1$ and $\xi_2$ are derived from $S_i$. For example, are they random non-overlapping subsets?
10.	Algorithm 3, Line 5 (and Definition 5): Clarify which distance is being computed. Is this the optimal transport (OT) distance between the two sets of latent space representation?
11.	In the Experiments section, clarify which architecture was utilized for the set-based neural network in the pre-trained utility model. Page 3 mentions options such as "Set Transformer" or "Deep Sets".
12.	In Figure 2 (c), there appears to be a discrepancy between the budget range mentioned in the text and within the figure.
13.	In the subsection "Class Imbalance" on page 8, it is unclear what the class imbalance was in the original data and after the removal of the labels. Please add these details.

Minor comment: Page 6: “While numerous AL methods has been proposed” => have been proposed

**Q9 Complying With Reviewing Instructions:**

Yes

---

> ### Author Rebuttal · Authors · 2024-04-06
>
> We are greatly encouraged by the reviewer's positive feedback regarding the practicality and importance of our work. We are grateful for your detailed review and provide the following responses:
>
> > Specifically, it's unclear how OT can be leveraged to predict validation set accuracy. Is it because a smaller OT distance between two samples implies that their induced validation set accuracy will be similar as well?
>
> Thanks for your detailed question. The terminology of OT distance refers to the class-wise Wasserstein distance between training dataset and validation dataset [1]. The OT distance characterizes the upper bound of the validation performance of any given model under certain Lipschitz conditions. In our setting, we view OT distance as an effective signal for enhancing the reliability of utility value prediction.
>
> > Figure 1: Please adjust the figure to clarify that both input set 1 and input set 2 are processed by the utility function
>
> Certainly! Given currently we cannot upload image, we will adjust figure for our revised version.
>
> > Algorithm 1: Please provide definitions for each argument in the input to ensure the algorithm is self-contained.
>
> $B$ denotes the labeling budget, $U_{0}$ denotes the initial unlabeled pool and $S_{0}$ for initial labeled pool. $\mathcal{X}$ denotes the ground set of all data points. $b$ means each iteration in the single acquisition stage. $M$ means the number of filtered unlabeled examples by Margin Sampling (i.e. $M$ unlabeled instances with lowest margin scores), which is mentioned in Section 4.1 (e). $n$ means the number of utility samples collected per iteration during the pretraining stage. $S_{val}$ means the validation set.
>
> > Algorithm 1, Line 5: Define k to avoid the need to search for definitions throughout the paper.
>
> Sure. We will incorporate your suggestion into our future version. Additionally, for a comprehensive understanding of the terminology and overall problem definition, we kindly direct your attention to Section 3, "Problem Statement," where full notations are detailed.
>
> > Algorithm 1, Line 14: Clarify the rationale for training \hat{u}_i.
>
> Certainly! We would need to update the shared representation $\phi(\cdot)$ to ensure updated calculation of optimal transport distance between two embedding spaces for better utility samples augmentations. Please refer to Page 6, column 1, 3rd line from bottom for detailed discussion.
>
> > Algorithm 3: Please include definitions for the input arguments to ensure clarity.
>
> We will add all definitions in future versions. $\hat{u}$ means the learned utility model and $j$ is the iteration of acquisition stage. $b$ denotes the batch size per iteration for acquisition stage. $S_{j}$ is acquired set at iteration $j$ of acquisition stage. $M$ is number of filtered unlabeled examples by Margin Sampling (i.e. M unlabeled instances with lowest margin scores), which is mentioned in Section 4.1 (e). $U_{j}$ denotes the unlabeled pool at iteration $j$ of acquisition stage.
>
> > Algorithm 3, Line 2: Correct the output notation
>
> Thanks for spotting out the typo. We will correct it to $D_{i + 1}$
>
> > Algorithm 3, Line 3: Ensure clarity by avoiding the use of $i$ as the loop variable, as it has already been utilized to annotate
>
> We will change it to another variable in the revised version.
>
> > Algorithm 3, Line 4: Describe how $\xi_{1}$ and $\xi_{2}$ are derived from $S_{i}$ . For example, are they random non-overlapping subsets?
>
> Sure. They are randomly sampled subsets. We do not require non-overlapping property, as we are not calculating any distance between $\phi(\xi_{1})$ and $\phi(\xi_{2})$. All we need is to augment high-quality utility samples by obtaining nearly accurate utility value $u_{1}$ and $u_{2}$.
>
> > Algorithm 3, Line 5 (and Definition 5): Clarify which distance is being computed. Is this the optimal transport (OT) distance between the two sets of latent space representation?
>
> Yes. We calculate the OT distance by utilizing utility model latent space representation.
>
> > In the Experiments section, clarify which architecture was utilized for the set-based neural network in the pre-trained utility model.
>
> Certainly. We use DeepSets and please see the detailed discussion of network architecture in Appendix A.1 and A.2 for all four datasets.
> > In Figure 2 (c), there appears to be a discrepancy between the budget range mentioned in the text and within the figure.
>
> Thanks for spotting out the typo! The labeling budget is 5000, 6000, 7000, 8000, 9000, 10000.
>
> > In the subsection "Class Imbalance" on page 8, it is unclear what the class imbalance was in the original data and after the removal of the labels. Please add these details.
>
> The class-imbalanced refers to the setting where 20% of groundtruth labels of training data are flipped to other remaining labels.
>
> [1] Just, Hoang Anh, et al. "Lava: Data valuation without pre-specified learning algorithms." arXiv preprint arXiv:2305.00054(2023).

---

### Official Review · Reviewer_LsER · 2024-03-20

**Q2-1 Originality-Novelty:** 3
**Q2-2 Correctness-Technical Quality:** 3
**Q2-5 Clarity Of Writing:** 3

**Q1 Summary And Contributions:**

Nowadays, most active learning methods tend to adopt costly multi-round interactive annotation strategies and existing state-of-the-art methods can be suboptimal in single-round selection. This paper presents a new active learning method called RAMBO, which adopts strategies such as bilevel optimization and greedy margin sampling. The method is further experimentally validated and shows that RAMBO is well-performed and robust.

**Q2-3 Extent To Which Claims Are Supported By Evidence:**

2: Fair: the main claims are somewhat supported by evidence (but the experimental evaluation may be weak, or does not match entirely with the claims, important baselines may be missing, proofs contain important ideas but lack rigor, algorithmic details are only discussed superficially, references are imprecise, assumptions are not sufficiently motivated or explicated, etc.).

**Q2-4 Reproducibility:**

2: Fair: key resources (e.g. proofs, code, data) are unavailable but key details (e.g. proof sketches, experimental setup) are sufficiently well-described for an expert to confidently reproduce the main results.

**Q3 Main Strengths:**

1.	The paper is well motivated and the proposed method effectively addresses the shortcomings of existing approaches in one-round-interaction problem.
2.	The strategies employed in the paper are well-founded and rational. The train of thought in proposing the method is clear.
3.	For the most part, this paper is well-written, and is easy to digest for someone who is familiar with active learning.

**Q4 Main Weakness:**

The main weaknesses of the paper are concentrated in the experimental aspect. See detailed comments.

**Q5 Detailed Comments To The Authors:**

1.	In Figure 2 (a), when there is a significant increase in labeling budget (from 500 to 700), RAMBO's performance actually exhibits a notable decline. The reasons behind need to be explained in detail.
2.	In the ablation study on $λ_{OT}$, the value of $λ_{OT}$ varys only within a small range, rather than spanning multiple orders of magnitude, which leads to failure to effectively illustrate the relationship between accuracy and parameter variations. The authors could consider values like 0, 10e-2, 10e-1, 1, 10, 100.
3.	In the ablation study of bi-level training, OT distance and RankNet, different combinations may result in performance variations. However, the reasons for the performance variations have not been thoroughly analyzed.
4.	Experiments in section 5.3 Indicate that RAMBO exhibits good robustness. However, the specific design aspects of the model that contribute to its robustness need to be elaborated in this section.

**Q9 Complying With Reviewing Instructions:**

Yes

---

> ### Author Rebuttal · Authors · 2024-04-06
>
> Thank you for the constructive feedback. We are pleased to hear that you find our paper is well-motivated and technically sound.
>
> > In Figure 2 (a), when there is a significant increase in labeling budget (from 500 to 700), RAMBO's performance actually exhibits a notable decline. The reasons behind need to be explained in detail.
>
> Thank you for the thorough discussion. MNIST is a relatively easy dataset, where nearly all baselines achieve a downstream accuracy of over 93% with 700 labeled instances. We respectfully highlight that the observed reduction in validation set accuracy from 96% to 95.5% remains within an acceptable range. This slight variation does not signify a noteworthy decline.
> It's worth noting that Figures 5(b) and 6(b) in Appendix B.2 represent noisy and imbalanced variations of the MNIST dataset. Under these scenarios, the previously observed decline in validation accuracy when expanding the labeling budget from 500 to 700 does not persist. Viewing these noisy/imbalanced versions as variations of the clean dataset, we see that once MNIST is subjected to perturbation, the decline in validation set accuracy disappears. This supports our view that the decline isn't a consistent phenomenon across all scenarios, even within the MNIST dataset itself, let alone across the other three datasets under consideration.
>
> > In the ablation study on \lambda_OT, the value varys only within a small range, rather than spanning multiple orders of magnitude, which leads to failure to effectively illustrate the relationship between accuracy and parameter variations. The authors could consider values like 0, 10e-2, 10e-1, 1, 10, 100.
>
> Certainly! Please see the table below with CIFAR10 as an example dataset. Each value represents average validation accuracy and the parentheses denote standard error.
>
> | Labeling Budget \ $\lambda_{OT}$ | 0      | 10e-2   | 10e-1   | 1      | 10     | 100    |
> |-----------------|--------|---------|---------|--------|--------|--------|
> | 3000            | 0.714(0.002) | 0.718(0.006) | 0.728(0.003) | 0.721(0.002) | 0.724(0.003) | 0.718(0.005) |
> | 5000            | 0.752(0.003) | 0.761(0.004) | 0.760(0.003) | 0.772(0.002) | 0.755(0.003) | 0.757(0.004) |
> | 7000            | 0.771(0.003) | 0.774(0.004) | 0.779(0.005) | 0.778(0.003) | 0.779(0.003) | 0.788(0.009) |
> | 10000           | 0.793(0.003) | 0.800(0.003) | 0.801(0.004) | 0.809(0.002) | 0.803(0.002) | 0.816(0.005) |
>
> > In the ablation study of bi-level training, OT distance and RankNet, different combinations may result in performance variations. However, the reasons for the performance variations have not been thoroughly analyzed.
>
> Table 1 underscores the non-separability of these modules, illustrating that configurations employing a bilevel approach consistently outperform non-bilevel setups, as evidenced by the comparisons in line 2 versus line 6 of Table 1, and across all entries in Table 4 of Appendix B.3. By considering the data presented in Table 1 of the main paper along with Tables 2, 3, and 4 in Appendix B.3, our objective is to demonstrate that omitting any single module leads to a reduction in performance compared to the integration of all three modules. This comprehensive approach aims to enhance the overall efficacy of the utility model, underscoring the integral role of each component in achieving optimal outcomes.
>
> > Experiments in section 5.3 Indicate that RAMBO exhibits good robustness. However, the specific design aspects of the model that contribute to its robustness need to be elaborated in this section.
>
> Thanks for your helpful suggestion. We aim to strengthen that
>
> **1. More robust objective**
>
> Ranknet transforms predicting the exact utility value of individual samples of data points (i.e. regression) to comparing which utility value is larger given two samples of data points with equal size (i.e. ranking). Prior experimental work on other domains, such as in language domains [1], utilizes the idea of asking labelers to rank outputs of prompts for reward model training. Here, we view the utility model as labelers and simplify our learning task from ranking multiple outputs of prompts to ranking utility value of two utility samples.
>
> **2. More robust inputs**
>
> We have incorporated a data augmentation strategy into our framework, specifically adapting the interpolation consistency regularization strategy detailed in Section 4.2.2. This approach is employed for artificially generating utility samples. The validation accuracy, which serves as a measure of utility, remains consistent across various sizes of validation sets (Figure 3(c)). Consequently, we generate utility samples by linearly interpolating between the utility values of two samples, along with their optimal transport (OT) distance, as outlined in Definition 5, Section 4.2.2.
>
> [1] Ouyang, Long, et al. "Training language models to follow instructions with human feedback." Advances in neural information processing systems 35 (2022): 27730-27744.

---

### Official Review · Reviewer_Ee21 · 2024-03-21

**Q2-1 Originality-Novelty:** 2
**Q2-2 Correctness-Technical Quality:** 3
**Q2-5 Clarity Of Writing:** 3

**Q1 Summary And Contributions:**

This paper introduces RAMBO (Ranking-based Active Learning via Multitask Bilevel Optimization), a one-round active learning method designed to efficiently select informative batches of unlabeled data for improving model performance. It breaks the limitations of traditional multi-round active learning by employing a ranking-based strategy within a multitask bilevel optimization framework, which optimizes the selection of data batches based on their predicted utility.

**Q2-3 Extent To Which Claims Are Supported By Evidence:**

2: Fair: the main claims are somewhat supported by evidence (but the experimental evaluation may be weak, or does not match entirely with the claims, important baselines may be missing, proofs contain important ideas but lack rigor, algorithmic details are only discussed superficially, references are imprecise, assumptions are not sufficiently motivated or explicated, etc.).

**Q2-4 Reproducibility:**

2: Fair: key resources (e.g. proofs, code, data) are unavailable but key details (e.g. proof sketches, experimental setup) are sufficiently well-described for an expert to confidently reproduce the main results.

**Q3 Main Strengths:**

1. This work is cost-effective, by requiring only one round of interaction with annotators, RAMBO reduces the overall cost and time associated with the labeling process.

2. RAMBO allows for efficient utility estimation of data points, minimizing the need for actual label acquisition.

**Q4 Main Weakness:**

1. This model relies on validation accuracy as a significant metric for determining the utility of data samples. But how to get a large enough validation set in scenarios where labeled data is scarce?

2. Too many hyperparameters, like those for the OT distance loss and bi-level optimization should be included in this work.

3. Sometimes RAMBO cannot get the optimal performance compared with baselines, e.g., Figure 2D.

**Q5 Detailed Comments To The Authors:**

1. Why does DULO always have the same trend with random sampling?

2. Why different classifiers are selected for different tasks? E.g., LeNet and ResNet-18, why not using the unified classifier?

3. In Table 1, it seems that With bilevel settings, the performance even worse. (See line 4 with accuracy 70.5%)

**Q9 Complying With Reviewing Instructions:**

Yes

---

> ### Author Rebuttal · Authors · 2024-04-06
>
> Your thoughtful discussion is greatly appreciated. We were pleased to hear that you found our algorithm is cost-effective while serving as an efficient estimation of utility of data points. Here we clarify your primary concerns and will incorporate the discussion for next version.
>
> **1. Large validation set size vs. scarce labeled data**
>
> Please see the table below as a sensitivity analysis of validation accuracy w.r.t the size of the validation set (averaged across 10 trials). In Figure 3(c), we examine the average validation accuracy versus validation set size across 100 randomly collected utility samples. These results demonstrate the robustness of validation accuracy as a consistent measure of utility value, the standard error inside the parentheses generally decreasing as the validation set size increases. Even at smaller validation set sizes of 50 or 100—which are significantly less than the pretraining set size $\mathcal{S}_0=k$—the accuracy measures are comparable to those observed with larger sizes, such as 800 or 1000.
>
> | Dataset\validation size | 50           | 100          | 200          | 400          | 600          | 800          | 1000         |
> |-------------------------|--------------|--------------|--------------|--------------|--------------|--------------|--------------|
> | SVHN                    | 0.132(0.018) | 0.132(0.012) | 0.133(0.011) | 0.130(0.010) | 0.136(0.009) | 0.130(0.009) | 0.130(0.008) |
> | MNIST                   | 0.758(0.007) | 0.757(0.005) | 0.763(0.001) | 0.764(0.003) | 0.760(0.009) | 0.770(0.009) | 0.764(0.008) |
> | FashionMNIST            | 0.699(0.023) | 0.714(0.015) | 0.707(0.011) | 0.712(0.009) | 0.712(0.009) | 0.707(0.009) | 0.712(0.009) |
> | CIFAR10                 | 0.358(0.020) | 0.372(0.016) | 0.370(0.007) | 0.365(0.008) | 0.366(0.005) | 0.368(0.003) | 0.366(0.003) |
>
>
> **2. Too many hyperparameters**
>
> Hyperparameters are reasonable concerns. However, a number of hyperparameters is a widely accepted practice in the design of learning-based acquisition functions, as detailed in the reference for LLAL. For $\lambda_{OT}$, Figure 2(f) and 7 (a)(b)(c) demonstrates the variability of $\lambda_{OT}$ across different acquisition budget.
> Please see the table for labeling budget vs. $\lambda_{OT}$ discussion
> in https://openreview.net/forum?id=o5Iw3kN9Eg&noteId=ikjSWfGAuz
> For the bi-level optimization (Section 4.2.1), we would like to believe bilevel-optimization process itself is crucial for regularizing utility functions to generalize to longer length utility samples [1,2].
>
> **3. Sometimes RAMBO cannot beat baselines, e.g., Figure 2D.**
>
> For detailed discussions regarding the SVHN dataset, please refer to paragraph 2 of Appendix B.2. While we recognize the unique characteristics of the SVHN dataset, it is important to clarify that RAMBO consistently ranks within the top two in terms of performance. LLAL lacks robustness across various datasets(See Figures 2(a) and (b)).
>
> **4. Same trend for DULO with random**
>
> Please refer Figure 2(b) for dataset FashionMNIST in the clean setting and Figure 5(d) 6(d) for dataset SVHN in noisy and class-imbalanced settings. Adding some perturbations by injecting some noise or artificially designing imbalanced classes, DULO shall exhibit much more variances than random baseline.
>
> **5. Two classifiers**
>
> Using ResNet-18 for MNIST and FashionMNIST might be an overkill as MNIST is a relatively simple dataset consisting of grayscale images of handwritten digits with a resolution of 28x28 pixels. ResNet18 is a complex architecture designed for much more challenging image recognition tasks. Moreover, due to its depth and complexity, ResNet18 has a lot more parameters compared to simpler models. Training such a large model on a 700 data points for MNIST could lead to overfitting with poor generalization. In fact, with 700 labeled MNIST data points, ResNet-18 structure only achieves average 78% validation set accuracy.
>
> **6. Bilevel settings might lead to poor performance (See line 4 with accuracy 70.5%)**
>
> We would like to reemphasize the synergistic importance of the three modules in question. Table 1 underscores the non-separability of these modules, illustrating that configurations employing a bilevel approach consistently outperform non-bilevel setups, as evidenced by the comparisons in line 2 versus line 6 of Table 1, and across all entries in Table 4 of Appendix B.3. By considering the data presented in Table 1 (main paper) with Tables 2, 3, and 4 in Appendix B.3, our objective is to demonstrate that omitting any single module leads to a reduction in performance compared to the integrated operation of all three modules.
>
> [1] Rajeswaran, Aravind, et al. "Meta-learning with implicit gradients." Advances in neural information processing systems 32 (2019).
>
> [2] Finn, Chelsea, Pieter Abbeel, and Sergey Levine. "Model-agnostic meta-learning for fast adaptation of deep networks." International conference on machine learning. PMLR, 2017.

---

### Official Review · Reviewer_HKTW · 2024-03-23

**Q2-1 Originality-Novelty:** 3
**Q2-2 Correctness-Technical Quality:** 3
**Q2-5 Clarity Of Writing:** 3

**Q1 Summary And Contributions:**

The paper introduces an active learning approach aimed at reducing labeling costs by selecting batches of unlabeled instances through a learned surrogate model for data acquisition. The key contributions are as follows:

1. Proposes a multi-task bilevel optimization framework that predicts the relative utility—measured by validation accuracy—of different training sets to ensure the learned acquisition function generalizes effectively.
2. Introduces efficient interpolation-based surrogate models to estimate the utility function, reducing the evaluation cost when validation accuracy is expensive to evaluate.
3. Demonstrates the effectiveness of the approach through extensive experiments on standard active classification benchmarks, showcasing performance improvements over existing methods.

**Q2-3 Extent To Which Claims Are Supported By Evidence:**

3: Good: the main claims are supported by convincing evidence (in the form of adequate experimental evaluation, proofs, (pseudo-)code, references, assumptions).

**Q2-4 Reproducibility:**

2: Fair: key resources (e.g. proofs, code, data) are unavailable but key details (e.g. proof sketches, experimental setup) are sufficiently well-described for an expert to confidently reproduce the main results.

**Q3 Main Strengths:**

1. The work differentiates from existing studies by proposing a learning-based approach that focuses on predicting the utility of unlabeled data using a novel framework that incorporates multi-task bilevel optimization.
2. The datasets used for evaluation (MNIST, FashionMNIST, CIFAR10, SVHN) are relevant, reliable, and have been widely used in similar studies.
3. Results are clearly presented and supported by data from extensive experiments.

**Q4 Main Weakness:**

1. The bilevel optimization framework, while effective, introduces computational complexity that might hinder scalability to extremely large datasets. This can potentially be a problem as active learning typically requires to be fast to be useful in real-world applications.
2. Given its reliance on a learned acquisition function, there's a risk of overfitting to the utility estimation, especially in data-scarce scenarios.
3. Relies on assumptions like the linear relationship between training data and model predictions, which might not hold across all scenarios.

**Q5 Detailed Comments To The Authors:**

na

**Q9 Complying With Reviewing Instructions:**

Yes

---

> ### Author Rebuttal · Authors · 2024-04-06
>
> We appreciate your insightful discussion. We were encouraged that you see our method “differentiating from existing studies by proposing a learning-based approach that focuses on predicting the utility of unlabeled data using a novel framework that incorporates multi-task bilevel optimization.” We address your detailed comments, which are helping us revise the paper and shape our future directions.
>
> > The bilevel optimization framework, while effective, introduces computational complexity that might hinder scalability to extremely large datasets. This can potentially be a problem as active learning typically requires to be fast to be useful in real-world applications.
>
> We acknowledge your concerns regarding the computational complexity and its potential impact on scalability. Furthermore, for a detailed runtime analysis across all four datasets, please refer to Appendix B.5. It is true that learning-based acquisition functions may require more pretraining time compared to heuristic-based ones. However, this trade-off between computational efficiency and model accuracy is often justifiable. In practical scenarios, the pretraining time is generally minimal compared to the duration of acquisition stage. Our experimental results demonstrate that RAMBO exhibits exceptional generalizability across various acquisition budgets. This makes it a valuable consideration for real-world applications.
>
>
> > Given its reliance on a learned acquisition function, there's a risk of overfitting to the utility estimation, especially in data-scarce scenarios.
>
> We greatly value your feedback and acknowledge the potential risk of overfitting in our utility estimation. We would like to highlight that in scenarios with a small k, RAMBO has demonstrated significant superiority over other baselines such as DULO and LLAL. Our approach parallels the challenges encountered in few-shot learning, where we similarly apply principles of meta-learning and pretraining within similar domains, as noted in [1]. Additionally, we enhance the utility value estimation by integrating a multi-task learning framework, which involves augmenting utility values through the interpolation of OT distances. Notably, [2] offers theoretical assurances that underscore the robustness of the multi-task learning framework, particularly against outlier tasks, which further mitigates the risk of overfitting.
>
> > Relies on assumptions like the linear relationship between training data and model predictions, which might not hold across all scenarios.
>
> Thank you for raising this concern. We would kindly point out that the related work datamodels [3] primarily relies on linear assumptions between training data and model predictions, as their methodologies fully utilize training data. In contrast, our focus is on active learning where we deal with limited labeled data. Consequently, we employ more expressive structures, such as set-based neural networks, to learn non-linear predictions from set-based embeddings, which are inherently non-linear. We do acknowledge, however, that the augmentation of utility samples in our model presupposes a linearity assumption (for details, please refer to Definition 5 in Section 4.2.2).
>
>
> [1] Rajeswaran, Aravind, et al. "Meta-learning with implicit gradients." Advances in neural information processing systems 32 (2019).
>
> [2] Duan, Yaqi, and Kaizheng Wang. "Adaptive and robust multi-task learning." The Annals of Statistics 51.5 (2023): 2015-2039.
>
> [3] Ilyas, Andrew, et al. "Datamodels: Predicting Predictions from Training Data." Proceedings of the 39th International Conference on Machine Learning. 2022.

---

### Meta-Review · Area_Chair_euhj · 2024-04-13

This paper proposes a new paradigm for active learning where a theory-motivated acquisition function is replaced by a learned surrogate function. The main challenge is to learn a surrogate function that generalizes well. The proposed approach is evaluated on standard machine learning benchmarks, such as MNIST and CIFAR-10.

This paper got positive reviews from all reviewers. While there are many comments, the reviewers also see the potential of learning acquisition functions for active learning.